Kinship identification using age transformation and Siamese network

Abbas Arshad 2016phdcs01@student.uet.edu.pk
Shoaib Muhammad
Department of Computer Science, University of Engineering and Technology , Lahore, Punjab , Pakistan
Asif Muhammad
Electronic publication date: 2022 Jun 8
Publication date: 2022
Volume: 8
Electronic Location ID: e987
Received 2022 Feb 9; Accepted 2022 Apr 29
Copyright: © 2022 Abbas & Shoaib
Copyright year: 2022
Copyright holder: Abbas
License: This is an open access article distributed under the terms of the Creative Commons Attribution License, which permits unrestricted use, distribution, reproduction and adaptation in any medium and for any purpose provided that it is properly attributed. For attribution, the original author(s), title, publication source (PeerJ Computer Science) and either DOI or URL of the article must be cited.
License URL: https://creativecommons.org/licenses/by/4.0/

Keywords: Kinship identification, Age transformation, Convolutional neural networks, Face encoding, Data mining & machine learning, Artificial intelligence, Data science, Social computing, Algorithms and analysis of algorithms, Computer education

Funding: The authors received no funding for this work.

==============================
Facial images are used for kinship verification. Traditional convolutional neural networks and transfer learning-based approaches are presently used for kinship identification. The transfer-learning approach is useful in many fields. However, it does not perform well in the identification of humans’ kinship because transfer-learning models are trained on a different type of data that is significantly different as compared to human face image data, a technique which may be able for kinship identification by comparing images of parents and their children with transformed age instead of comparing their actual images is required. In this article, a technique for kinship identification using a Siamese neural network and age transformation algorithm is proposed. The results are satisfactory as an overall accuracy of 76.38% has been achieved. Further work can be carried out to improve the accuracy by improving the Life Span Age Transformation (LAT) algorithm for kinship identification using facial images.

Introduction

The pictorial data generated by businesses, social media, public industry, non-profit sectors, and scientific research have increased tremendously (Jin et al., 2015). This graphical data contains much useful and worthwhile information that could be used for various purposes (Chen, Mao & Liu, 2014; Emani, Cullot & Nicolle, 2015).

In the last few years, researchers became interested in extracting kinship information from pictorial data with human faces, which can be used for different purposes. As face image data provides different unique features of humans and contains a wealth of information that can be used for various purposes (Zhou et al., 2012a). The purpose of extracting genetic relationships between human images is to verify human kinship, which is useful information for medical sciences, psychologists, security agencies and family album organizations. Furthermore, it can be utilized in image annotation, searching for missing children, human trafficking, and solving immigration and border patrol (Zhou et al., 2012a; Lu et al., 2013; Yan et al., 2014).

Face recognition and verification have been an active area of research for the last two decades. It has been studied enthusiastically to make computers capable as more intelligent applications have been developed for human-computer interface (HCI), security, robotics, entertainment and games, etc. (Bowyer, Chang & Flynn, 2006). In parallel, after face recognition, now age and gender detection techniques have also been proposed. For example, Levi & Hassner (2015) proposed a classification technique for age and gender using convolutional neural networks. Similarly, Dehghan et al. (2014) proposed the genetic identification technique in which they used gated autoencoders and tried to determine the resemblance between parent and child. They found the resemblance by using the father, mother and children's facial features similar to those found in anthropological studies. However, it only works if there is much resemblance of offspring with parents and therefore, it causes poor performance in case children have little or no resemblance with their parents. On the other hand, Yan & Hu (2018) proposed another technique of kinship verification for videos using the video face dataset KFVW, which was prepared in wild conditions to handle kinship verification for the video-based study. This technique addresses some pose identification. However, the experimental data indicates that matric-based learning is not an effective technique for kinship identification.

Amongst these challenges, researchers are adopting the pre-trained network convolutional neural network as a transfer learning with CNN layered architecture and training algorithm to get better results for kinship identification & verification. However, the shortcoming of such models is that they lack kinship identification as transfer learning models are trained on a significantly different type of data compared to human face images. Moving onwards, to solve the problems of limited datasets. Robinson et al. (2017) introduced a database named Recognizing Family In the Wild (RFIW). It is the first large-scale image database, especially for kinship recognition and exploits the challenges of kinship recognition.

Meanwhile, methodologies proposed so far have several challenges, such as limited pair of images for parents and children. Moreover, a classifier trained using transfer learning and a limited scale dataset fails in recognizing real-world images. In this regard, this study aims to propose and provide an effective technique for performing kinship identification through image data.

Instead of comparing direct images of parents and children, this research work suggests an approach of age transformation and converts images of parents and children into the same age of 15–19-year age and then compares them to get better accuracy of similarity. In this model, we have used a pre-processing stage of age transformation before image comparison for kinship identification and verification. At first, our model uses an age transformation algorithm to transform facial images by increasing or decreasing the age of face images and making them at the same age stage. After making images at the same stage of age, makes it easy to compare images and finding similarities between them to exploit kinship identification between them. Furthermore, the robustness of our technique is validated through extensive experiments and analysis on a huge dataset. Figure 1 shows a high level overview of the proposed methodology.

Figure 1 High level methodology of proposed study.

Diagram showing the operation of the methodology.

The contributions of this article include (1) proposing improved pre-processing of dataset images through employing the use of the Life Span Age Transformation (LAT) algorithm for transforming the images onto the same scale of age, (2) using the Siamese network for performing the feature extraction from the transformed images, (3) introduced technique is validated by using the state-of-the-art benchmarked dataset namely RFIW, (4) finally, extensive experiments conducted on the dataset using the proposed technique to identify the improved effectiveness. Moreover, the comparative analysis indicates that the proposed technique outperformed the existing methods.

Related work

Researchers are interested in kinship verification (family or not family) in the computer vision community by applying different face recognition and machine learning techniques. Fang et al. (2010) introduced the problem and used simple features for kinship identification like eyes and skin color and distances between facial parts for kinship verification. Subsequently, Xia et al. (2012) claimed the similarity between parents and their children is quite large and proposed an approach of kinship learning by removing the gap between two facial images of a parent, one image of young age and another image of old age and children’s images. Lu et al. (2014) used a metric learning approach for kinship verification and found effective features, which provided the most discriminative results. Levi & Hassner (2015) proposed a classification methodology using age and gender by applying convolutional neural networks and got better results. Dehghan et al. (2014) proposed the genetic identification technique by determining resemblance between parent and offspring via gated autoencoders. They used deep learning techniques to learn the most discriminative features between parents and children to find out their resemblance. That approach deals with resemblance by using the father and mother’s facial shapes and extracting a similar face with a combination of facial feathers of the father and mother (Dehghan et al., 2014). Yan & Hu (2018) revealed that Euclidian similarity metric is not a powerful way to measure the similarity of facial images, especially when captured in wild conditions. They clarified that the similarity metric can handle the problem better to deal with face variations compared to Euclidian similarity. They used a mid-level feature vector with discriminative metric learning and proposed a prototype-based feature learning approach for kinship verification (Lu et al., 2014; Yan et al., 2014). Yan & Hu (2018) proposed a methodology of video-based kinship verification by using data set of video faces called Kinship Face Videos in the Wild (KFVW). The dataset was built by capturing facial images from videos for kinship verification. This methodology analyzes the human faces in the video by getting training set from video poses and then applying distance metric learning approaches to get a positive semi definite matrix (PSD) for face recognition and kinship identification (Yan & Hu, 2018). Robinson et al. (2017) introduced the first large-scale image database for kinship recognition called Families In the Wild (FIW) and exploits the challenges in kinship recognition. The FIW database consists of thousands of images of faces for kinship recognition. Li et al. (2017) presented a framework in which knowledge of face recognition from large-scale data-driven transferred and then fine-tuned metric space to get discrimination of kin related people. They also proposed an augmented strategy to balance family members’ images and used triplet and ResNet to extract face encoding for kinship identification. In early techniques, kinship verification uses handcrafted descriptors from facial images to perform classification for learning. Fang et al. (2010) used facial features like eye and skin colors and distance of eye-to-nose for kinship verification. Zhou et al. (2012b) proposed an approach based on spatial pyramid features for kinship verification. This approach used Gabor-based facial image gradient orientation features. Liu et al. (2017) applied a transferrable approach of fisher vectors derived from each facial image to extract similarity for kinship verification (Robinson et al., 2018). Kohli, Singh & Vatsa (2012) proposed an approach to achieve kinship similarity using a self-similarity descriptor. They introduced that kinship verification is a two-factor classification problem. They revealed that low-level features could not be used as an underlying source of visual resemblance between people with kinship relations. In Shallow metric-based approaches, metric learning methods are used to learn discriminative features for kinship verification. These approaches learn a Mahalanobis distance using handcrafted features identification and get a better score of similarity between kinship-related pairs with non-kinship-related pairs (Lu et al., 2013). In the Deep learning-based approach, He et al. (2016) and Kohli et al. (2017) motivated kinship identification and verification after getting impressive success by applying deep learning approaches to classify different facial images. Many techniques have been adopted for deep metric learning to get discriminant features for kinship verification. Dehghan et al. (2014) introduced an approach of fusing the features using gated auto-encoders. They extracted optimal features by reflecting parent-offspring resemblance. Wang, Robinson & Fu (2017) proposed the Kinship Verification on Families in the Wild with Marginalized Designing Metric Learning (DML). That technique used the largest kinship verification using Auto-encoder and Discriminative Low-rank Metric Learning (DLML) algorithm for feature discrimination. After using matric learning, researchers found a better way to find similarity for kinship identification by using a convolutional neural network. Zhang et al. (2015) adopted an approach of kinship verification using a CNN to train the algorithm with concentrated image pairs. He et al. (2016) introduced a deep residual learning approach for image recognition. Their approach used residual training with neural networks and multiple layers as learning residual functions. Duan, Zhang & Zuo (2017) proposed a deep kinship verification technique named Coarse to Fine Transfer (CFT) using CNN from face recognition to kinship recognition and used deep transfer learning. Yan & Hu (2018) proposed a kinship verification technique, which works on videos. This technique uses distance metric learning on Kinship Face Videos in the Wild (KFVW) dataset for kinship verification. Lu, Hu & Tan (2017) developed a discriminative deep multi-metric learning (DDMML) methodology. They used multiple neural networks jointly to maximize the association of different features of each sample and reduce the distance of positive pair and increase the distance of negative pair. Li et al. (2017) introduced the kinship verification technique using KinNet: Fine-to-Coarse Deep Metric Learning and Pre-training the network and minimizing a soft triplet loss. They used four CNN networks to boost the performance. Liu et al. (2017) introduced SphereFace, a deep hyper sphere embedding for face recognition. They addressed angular SoftMax loss and angular margins problem. Their technique uses a 64-layer CNN neural network for training and discriminative constraints on a hypersphere to get a better face recognition (FR) problem under open-set protocol. Ozkan & Ozkan (2018) introduced an approach of synthesizing child faces with a pre-trained model by analyzing facial images of parents. Yan & Song (2021) worked on multi-scale deep relational reasoning for facial kinship verification and used two convolutional neural networks, which shared network parameters and extracted different scales of features for kinship identification. After using a convolutional neural network, researchers moved to find kinship using the Generative Adversarial Network (GAN), introduced by Goodfellow et al. (2020). Ghatas & Hemayed (2020) proposed GANKIN: generating kin faces using disentangled GAN and image synthesis approach from parents to children, they also used pertained FaceNet and GAN network. Nguyen, Nguyen & Dao (2020) proposed an approach of recognizing families through images with pertained encoders. They used pre-trained networks FaceNet, Siamese and FGG network to get face image encoding and find kinship between facial images. Keeping in view the efficiency factor of GAN based approaches, we also used GAN based age transformation algorithm and Siamese network to build and train our model.

Although some encouraging results have been obtained from proposed methodologies for kinship identification and verification in the last few years, automatic kinship verification is being performed poorly in the real-world applications used in daily life. Due to the non-availability of large-scale datasets, results are not too accurate to handle the kinship identification problems. Existing datasets like Family101, UB KinFace, Cornell KinFace, KinFaceW-I, and KinFaceW-II provide a few examples, but they fail to achieve accurate distributions of genetic or kinship relationships. Moreover, they have a limited pair of images for parents and children; Classifier trained on a limited scale dataset fails while recognizing real-world images.

To handle these issues, we proposed an approach to find the kinship relationship between parents and children. Our methodology uses age transformation and converts images of parents and children to the age of 15–20 because images of this age have maximum facial features, which can be a good source for the discrimination of features between facial images. After the process of age transformation and converting facial images to a young age for both parents and children, these faces get closer to each other in facial look and expression and then it makes it easy to find the similarity between them. With these images, there is much probability of getting parent’s faces and images close to each other. Ultimately, it will make it easy for the face encoder to generate close face encoding. As a result, we get a low distance value while finding cosine similarity. Figure 2 shows the effect of age transformation.

Figure 2 Effect of age transformation.

Diagram to elaborate effect of age transformation on facial images. Source credit: https://web.northeastern.edu/smilelab/fiw/.

Proposed work

This section outlines the proposed methodology for performing the kinship identification. In the proposed method, we presented a model of a deep relational network that uses a preprocessing stage of age transformation of two facial images before comparing them to exploit kinship relationships from facial images. This scheme first transforms facial images by increasing or decreasing the age factor and making two images into the same age stage and then compares them to find and verify kinship. After transforming facial images, we proposed the use of a Siamese network with two convolutional neural networks by sharing parameters between them. Afterward, it extracts different scales of features to find similarities between images by using triplet loss. We also aimed to conduct experiments on a widely used facial kinship dataset, namely RFIW. In this methodology, the proposed model uses age transformation and converts facial images at the same stage of age, between 15 and 19 years. However, we considered this age because, in this age period, a person’s face looks strong and can provide clear facial features and better encode facial images. Furthermore, after encoding transformed faces, we applied triplet loss on three faces of parents and images and extracted the kinship relationship between parents and images. In addition, we have employed parent’s images as anchor and negative part of the triplet while children’s images as a positive part of the triplet. We fixed the father and mother position of being positive or negative to each other while training in the Siamese network. Likewise, we used an age transformation algorithm that provided close pair of facial images of parents and children for processing to exploit kinship identification between them. This age transformation algorithm will provide images for processing to consider for kinship identification. More graphical representation and the working flow of our proposed methodology is depicted in Fig. 3.

Figure 3 Working of proposed methodology.

This diagram shows working of proposed methodology, how methodology works to achieve kinship identification. Source credit: https://web.northeastern.edu/smilelab/fiw/.

Model training

The proposed model uses age transformation and feature encoding of face images with triplet loss to extract facial similarity to identify kinship. The first stage converts facial images to images of persons having approximate ages of 15–19 years. After doing conversion of two input images with ages between 15–19 years, these converted images are processed with the Siamese network to extract feature encoding for further processing.

It uses ResNet 50 with two fully connected layers and one dense output layer to extract features. It extracts a feature vector of 128 × 128 for all input facial images and uses triplet loss to discriminate features for kinship identification. It maximizes the distance of the anchor image with a negative image and minimizes the distance with a positive image. The size of input images is 224 × 224 × 3 and the feature vector returned by the Siamese network is 128. During the training process, hard sample selection for positive or negative pairs are not equally important. The pairs with higher loss might have more impact on the model training. The training set can be defined as: Let Xa, Xp and Xn are finite set of images for Father, Children and Mother having ‘m’ number of images for each set.

(1) Xa={x1a,x2a,…xma}

where Xa is set of anchor images for father images.

(2) Xp={x1p,x2p,…xmp}

where Xp is set of positive images which are taken from children’s images.

(3) Xn={x1n,x2n,…xmn}

Xn is a set of negative images taken from the set of mother images.

Then input sample taken from these three sets will be a powerful set of three sets to make a set of triplets let X is the power of Xa, Xp and Xn, set then we get set X as a set of the triplet.

(4) X={(x1a,x1p,x1n),(x2a,x3p,x4n)…(xna,xnp,xnn)}

X is a power set of images having three members as triplet of anchor as xa, positive as xp and negative image as xn respectively where sequence of triplet members are anchor, positive and negative members with images of father, child, and mother respectively. After getting feature extracted from pertained Siamese network, we get a set of features:

(5) F(X)={[f(x1a),f(x1p),f(x1n)],{[f(x2a),f(x2p),f(x2n)]…{[f(xna),f(xnp),f(xnn)]}.

This sequence of the set is used for extracting similarities of children with father and mother to get kinship relation of Father-Son (F-S), Father-Daughter (F-D), Mother-Son (M-S) and Mother-Daughter. For sibling relationships, we changed some sequence of power set. We took one sibling image as an anchor, one as positive, and one as negative if the third image of the sibling did not exist. For negative position, we took any random image from the set of mother or father. So, for negative position random set of images: Xr = P {Xa | Xn}. Then set of triplets for sibling relationship Brother-Brother (B-B), Sister-Sister (S-S) and Brother-Sister (B-S) is as follows:

(6) Xs={(x1a,x2p,x1r),(x2a,x3p,x2r)…(xna,xmp,xnr)}

where Xs is a power set of images having three members as triplet of anchor as xa, positive as xp and negative image as xr respectively.

Loss function

The loss function for the triplet loss on the extracted feature, For three cases

1. While comparing the father’s image with the child’s image, if Df is distance of child’s image with father’s image and Dm is distance of child’s image with mother’s image then we define the loss function as:

(7) Df=||f(P)−f(A)||2,Dm=||f(P)−f(N)||2

and some margin ‘m’ as hyperparameter, whereas A, P, N are anchor, positive and negative images, and f(A), f(P) and f(N) are features of father, child, and mother respectively. If father’s image is closer to child’s image then we increase the distance of child’s image with mother’s image and decrease the distance of child’s image with father’s image, so loss function to get similarity between father and child will be:

(8) £f(A,P,N)=max(Df−Dm+m,0)ifDf<Dm

2. While checking the similarity of children with mothers then, we revert the loss function. To find the similarity of the child image with the mother image, we increase the distance of child image with father image and decrease the distance of child image with mother image then loss function will be:

(9) £m(A,P,N)=max(Dm−Df+m,0)ifDf>Dm

3. While comparing images of siblings, we use distance measures of images of two siblings, S1, S2. We find the distance between siblings and random images as:

(10) Ds1=||f(S1)−f(S2)||2,Ds2=||f(S1)−f(Nr)||2

where f(S1), f(S2) and f(Nr) features of siblings and a random image, respectively. After calculating distance and using margin ‘m’ as hyper parameter, we can define the loss function as:

(11) £s(A,P,N)=Max(Ds1–Ds2+m,0)

Ds1 is the distance between one sibling with other siblings. Similarly, Ds2 is the distance of siblings with a random image to find triplet loss and minimize the distance between the first and second siblings.

Network structure

To select information from different scales of features for input to the relational network, we use the pre-trained Siamese network and get a feature map of size R512×1. Network contains three dense layers to down sample the features map and get a features vector of size 128 × 1. Each features vector of size 128 × 1 will provide information of the faces as face encoding. This face encoding is then used to find the cosine similarity between face images respectively. After that relational network analyzes these selected features with multi-layer perceptrons which consists of some fully connected layers and relu activation functions.

Following are steps of model training: Pictorial data is fetched from the data set and all images are converted to the same stage of age by LATS. After age transformation, an intermediate dataset is prepared for training from original images.

Transformed data is fetched into three vectors: father, mother and children, to prepare a triplet for the Siamese network

One vector is used as positive, one for negative and one as anchor

The triplet is used by the Siamese network to extract face features

Define the triplet loss function. It decreases the distance between positive and anchor images and increases the distance between positive and negative images.

Setting up for training and evaluation

This multi-layer perceptron will extract the relation of features and output feature of size R128×1. Then we compare these features of size R128×1 at the element level to represent the distance between features of faces.

Lastly, we use another multi-layer perceptron to find the similarity of faces for kinship identification from the relation of different face images. It also consists of some fully connected layers and relu activation functions.

A flow of model training is represented in Fig. 4.

Figure 4 Flow of model training with age transformation and face encoding.

This diagram shows flow of model training with age transformation and face encoding. Source credit: https://web.northeastern.edu/smilelab/fiw/.

The CNN structure uses Siamese network; its input size is 3 * 256 * 256 and final output features vector size is 128 * 1. This network has three dense layers subsequently with batch normalization and relu activation function to minizise the size of feature vector.

The relational network has three convolutional layers, each layer uses 128 feature vector of 10 images with batch normalization and relu activation function. The input feature size of each layer is R10×128×128×128 and last dense layer has output feature size of 1 × 128. It applies segmoid function to establish kin relationship between images, detailed relational network with input parameters is depicted in Table 1.

Table 1 Parameters of deep relational network.

Setting	Input size	Output size	Kernel	Stride	Padding	
SIAMESE NETWORK	3 * 256 * 256	512 * 1	3	1	0	
DENSE-1+ BN + RELU	512 * 1	256 * 1	2	2	0	
DENSE-2 + BN + RELU	256 * 1	256 * 1	2	1	0	
DENSE-3	256 * 1	128 * 1	2	2	0	
Relational Network	
Conv-1+ BN + RELU	10 × 128 × 128 × 128	3	1	2	
Conv-2+ BN + RELU	10 × 128 × 128 × 128	3	1	2	
Conv-3+ BN + RELU	10 × 128 × 128 × 128	3	1	2	
Dense (Flatten) + SIGMOID	1 × 128	Kin/Not Kin	

To optimize the network, contrastive loss function is used with below specifications:

(12) L(d,Y)=∑Ni[1/2∗Yi∗di2+(1–yi)∗1/2∗max(0,m–di)]

where L denotes the loss, N represents the number of samples, yi is the ground truth of ith sample, and di is the distance between the output of the encoder, m is margin parameter.

Similar face images are pushed close and dissimilar images pushed away to get maximum similarity between similar images.

Data set

We used a dataset of RFIW and took images of 200 families having good resolution images as a constraint of Life Span Age Transformation (LATS) that requires images having good resolution. LATS generates ten age clusters and each age cluster has ten images. We picked one age cluster between 15 and 19, so we used 10 images to train our model. We used images of ages between 15 and 19 years because in this age period person’s face looks strong and can provide clear facial features and we can get the better encoding of facial images. For model training, we used images of 200 families; each family has average four members. For each member we used 10 transformed images and our model is trained on approximate 200 × 4 × 10 = 8,000, from this pool of data, we used 30% data for validation.

As we used the LATS model for preprocessing and Siamese Network for training our model, which are CNN based network architecture, therefore we also adopted CNN network for feature extraction and training model to get compatibility with the existing model and achieve efficient results. Moreover, for high dimensional data CNN provides automatic feature extraction and forwards extracted features to the classifier to get classification results.

Results and Discussion

The CNN-based deep relational network is utilized for extracting the features from the facial images of the dataset. Table 1 outlines the details of the included parameters for the CNN-based deep relational network. Unlike the previously existing models, it represents that our model explicitly establishes relations between three feature maps rather than making relations within one another. Additionally, it depicts that our model takes ten images of each member and finds the triplet loss on 128 features maps of each ten images for one member. In total used, 30 features map for one comparison to find the similarity between them. The proposed model delivers the optimal performance by utilizing this methodology.

In this section of the study, we have listed the experiments and achieved results by employing the use of the proposed technique of utilizing a deep relational network along with the LAT age transformation algorithm (Or-El et al., 2020). We have used the large dataset of Recognizing family in the wild (RFIW) for the training and validation of our proposed technique. In the first phase, we converted images of datasets RFIW to different life stages for age transformation. After the age transformation of facial images, we converted images at the same stage of ages by adjusting the age factor. In the first stage, we transformed facial images by increasing or decreasing the age factor and making two images into the same stage of age. In the second phase, we trained our algorithm by comparing two images and evaluating metrics and parameter settings to extract kinship relation accuracy.

Age transformation

For age transformation, we employed the Lifespan Age Transformation Synthesis algorithm, proposed by Or-El et al. (2020). Using this algorithm, we prepared our data set images for comparison that converts images at different stages of life. Afterward, we picked the images of ages between 15–19 and used them for feature extraction. Table 2 outlines the training and validation accuracies observed in different relationships by utilizing the proposed model.

Table 2 Achieved model performance for different relationships.

Training accuracy	Validation accuracy	
Relation	Accuracy (%)	Relation	Accuracy (%)	Mean (%)	
Father-Son	80.12	Father-Son	76.16	78.14	
Father-Daughter	77.15	Father-Daughter	73.00	75.07	
Mother-Son	75.75	Mother-Son	72.75	74.25	
Mother-Daughter	79.17	Mother-Daughter	77.0	78.08	
Overall Mean	76.38	

Similarly, Table 3 represents the observed results on the baseline dataset. While comparing accuracy with a model trained on dataset RFIW, the results from Table 3 indicate that our proposed model has delivered better performance than the existing state-of-the-art models by improving the overall accuracy.

Table 3 Comparison of results on baseline dataset with other models.

Sr. No.	Methodology	Classifier	Accuracy (%)	
1	Robinson et al. (2018).
Visual Kinship Recognition of Families in the Wild	Cosine similarity
SVM	69.18	
2	Ghatas & Hemayed (2020). GANKIN: generating Kin faces using disentangled GAN	GAN	71.16	
3	Nguyen, Nguyen & Dao (2020). Recognizing Families through Images with Pretrained Encoder	Pre-trained CNN	73.21	
4	Proposed Methodology	Pre-trained LATS + Siamese	76.38	

Meanwhile, the previous models have failed to deliver improved performance for up to 73.21% accuracy. On the other hand, the proposed model has outperformed existing state-of-the-art models by delivering an accuracy of 76.38%. Furthermore, we plan to improve the model and accuracy in the future by improving the underlying relational network and applying it to transformed images with the same stage of age.

The major contribution of our research is to introduce a robust way of kinship identification by comparing images of parents and their children with transformed ages instead of comparing their actual images. Improved accuracy of methodology proved that we could get better results for kinship identification if we compare images after age transformation instead of comparing direct actual images. From the results obtained after training indicates that similarity between the same genders is greater than opposite gender because the similarity score between father-son and mother-daughter is greater than father-daughter and mother-son, respectively. The obtained results show that due to the same gender factor, daughter looks more similar to the mother compared to the father. Similarly, the son seems more similar to father rather than the mother.

Conclusion

Kinship identification is used for kinship verification by using facial images. Meanwhile, the previous studies have explored this area by employing transfer learning-based solutions. This study, however, presents a different approach to perform kinship verification.

In this study, we have introduced a technique that uses a pre-trained LAT model along with a Siamese network for performing kinship identification. Additionally, we have employed the age transformation approach to find similarities between parents with children. The extensive experimental results were used to validate the performance of our proposed model. Furthermore, the comparative analysis with previously carried out studies reflects that our model outperformed the existing state-of-the-art models using a similar approach, thereby delivering an overall accuracy of 76.38%. In the future, we aim to improve the model performance by improving the underlying relational network and applying it on transformed images with the same age stage.

Additional Information and Declarations

Competing Interests

Author Contributions

Data Availability

The authors declare that they have no competing interests.

Arshad Abbas conceived and designed the experiments, performed the experiments, analyzed the data, performed the computation work, prepared figures and/or tables, and approved the final draft.

Muhammad Shoaib conceived and designed the experiments, analyzed the data, authored or reviewed drafts of the article, and approved the final draft.

The following information was supplied regarding data availability:

The code and model are available at figshare: abbas, arshad (2022): Kinship Identification using LATS. figshare. Software. https://doi.org/10.6084/m9.figshare.19144349.v2

The third-party data is available from the Families In the Wild (FIW) Dataset: https://web.northeastern.edu/smilelab/fiw/. Contact Joe Robinson (robinson.jo@husky.neu.edu).

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
