# Peer review of "Kinship identification using age transformation and Siamese network"

_PeerJ Computer Science, doi:10.7717/peerj-cs.987_

## Round 0.1 · original submission · Minor Revisions

Dear authors please carefully revise the paper according to reviewer comments and resubmit

Reviewer 1 ·

Basic reporting

- Problem statement, solution provided and outcome should be clearly explained in abstract. The flow of information should be in order and connected.
- Description of big data in the beginning of introduction is not relevant and should be removed. Start from pictorial discussion.
- Clearly explain the problem statement in the introduction section.
- Contribution of paper mainly highlights the methodology not the contribution. Explain how the results and outcome of this research contributes to the body of knowledge.
- Remove paper organization section.
- "Joseph P.R. et al Introduced....." use standard style of reference citation in the text.
- "resemblance [8]. Zhang et al. (2015)" - follow only one style of referencing throughout the paper.
- Literature should be written in discussion form not simple descriptive form.
- "we present a model....." don't use I, we, our in research. use passive voice sentences.

Experimental design

In proposed work, use modeling form (graphical/mathematical etc) to explain the methodology and explain all the steps in sequence. Give research model.
- Support the methodology with references.
- Provide details of data set, its source and how much data is used for training and implementation?
- Equations are inline with text. Should be on separate lines, with standard equation format and numbering, and explain them.
- Properly define the design of methodology and explain it step by step.

Validity of the findings

- How the validity of the results checked?
- Explain results and show them with the proper technical format rather than simply mentioning them.
- Use tables, data, numbers and figures to explain the results.
- Data given in tables should be well explained and justified.
- How results are significant?
- Conclusion should explain how results support the objectives. what is the significance of the study. Summarize the outcome of the research, its impact and applicability.

Additional comments

- Make grammatical corrections.

Reviewer 2 ·

Basic reporting

no comment

Experimental design

a)-
What was the specific reason for picking images of an age between 15-20 only?
b)-
What was the dataset size used in the experimental validation.
c)-
Why CNN-based deep relational network was used for features extraction from the facial images of the dataset?

Validity of the findings

a)-
Why the proposed model is taking 10 images of each member? why not 5 or 15 or more?

b)-
Any specific reason why Father-Daughter and Mother-Son accuracy in both Training and Validation Accuracy is lower as compared to Father-Son and Mother-Daughter?

c)-
Have we also compared any other performance factor other than accuracy from the other proposed models mentioned in Table1.

Additional comments

We can only say that the proposed model's performance is better than the previously proposed model (Table1). We cannot say that the previously proposed models failed to deliver improved performance.

Reviewer 3 ·

Basic reporting

English language corrections are needed at some places in the article. Proofreading is recommended. E.g. line 194: Spelling mistake ‘pre-processing’, Line 284: use figure 4 instead of diagram 4

Experimental design

no comments

Validity of the findings

no comment

Additional comments

Repetition of the basic working is observed in multiple sections. It is suggested that more details be added instead of repetitions of the basic working.
Theme of Figure 3 and figure 4 is the same.
Care must be taken when using abbreviations. It is suggested that the full form and its abbreviation be provided at the first appearance. Afterward only abbreviated form be used. E.g. RFIW –Full form and its abbreviation should be provided at first appearance only. Afterward only abbreviated form should be used. Lines 62, 82, 315. Similarly, at line 324 only the abbreviation LAT should have been used. Many other such examples are present.
All equations must be numbered and cited in the text.

---

## Round 0.2 · accepted · Accept

Thanks for addressing the comments.